# The inverse association of state cannabis vaping prevalence with the e-cigarette or vaping product-use associated lung injury

Ellen Boakye[1,2], Omar El Shahawy[2,3], Olufunmilayo Obisesan[4], Omar Dzaye[1], Albert D. Osei[4], John Erhabor[1,2], S. M. Iftekhar Uddin[5], Michael J. Blaha[1,2] *

1 Johns Hopkins Ciccarone Center for the Prevention of Cardiovascular Disease, Baltimore, MD, United States of America, 2 The American Heart Association Tobacco Regulation and Addiction Center, Dallas, TX, United States of America, 3 Department of Population Health, New York University School of Medicine, New York, NY, United States of America, 4 Department of Medicine, MedStar Union Memorial Hospital, Baltimore, MD, United States of America, 5 Department of Medicine, Brookdale University Hospital Medical Center, Brooklyn, NY, United States of America

* mblaha1@jhmi.edu

**Data Availability Statement:** The data underlying the results presented in the study are available from https://www.cdc.gov/brfss/annual_data/

## Abstract

The e-cigarette or vaping product-use-associated lung injury (EVALI) epidemic was primarily associated with the use of e-cigarettes containing tetrahydrocannabinol (THC)- the principal psychoactive substance in cannabis, and vitamin-E-acetate- an additive sometimes used in informally sourced THC-containing e-liquids. EVALI case burden varied across states, but it is unclear whether this was associated with state-level cannabis vaping prevalence. We, therefore, used linear regression models to assess the cross-sectional association between state-level cannabis vaping prevalence (obtained from the 2019 behavioral Risk Factor Surveillance System) and EVALI case burden (obtained from the Centers for Disease Control and Prevention) adjusted for state cannabis policies. Cannabis vaping prevalence ranged from 1.14%(95%CI, 0.61%-2.12%) in Wyoming to 3.11%(95%CI, 2.16%-4.44%) in New Hampshire. EVALI cases per million population ranged from 1.90(0.38–3.42) in Oklahoma to 59.10(19.70–96.53) in North Dakota. There was no significant positive association but an inverse association between state cannabis vaping prevalence and EVALI case burden (Coefficient, -18.6; 95%CI, -37.5–0.4; p-value, 0.05). Thus, state-level cannabis vaping prevalence was not positively associated with EVALI prevalence, suggesting that there may not be a simple direct link between state cannabis vaping prevalence and EVALI cases, but rather the relationship is likely more nuanced and possibly reflective of access to informal sources of THC-containing e-cigarettes.

## Introduction

The e-cigarette or vaping product-use-associated lung injury (EVALI) epidemic in the United States (US) was an acute lung injury seen in persons who reported e-cigarette use before the onset of symptoms, many of whom were otherwise healthy [1, 2]. A majority of EVALI patients were young (median age, 24 years) and reported vaping e-liquids containing

annual_2019.html https://www.cdc.gov/tobacco/
basic_information/e-cigarettes/severe-lung-
disease.html.

**Funding:** Research reported in this publication was
supported by the National Heart, Lung, and Blood
Institute of the National Institutes of Health under
Award Number U54HL120163 (https://govtribe.
com/award/federal-grant-award/cooperative-
agreement-u54hl120163) granted to the American
Heart Association Tobacco Regulation and
Addiction Center. However, the content is solely the
responsibility of the authors and does not
necessarily represent the official views of the
National Institutes of Health. The funders had no
role in study design, data collection and analysis,
decision to publish, or preparation of the
manuscript.

**Competing interests:** The authors have declared
that no competing interests exist.

tetrahydrocannabinol (THC)–the principal psychoactive substance in cannabis, and vitamin-E acetate, an additive sometimes used in informally sourced THC-containing e-liquids, was found to be strongly linked to the epidemic [1].

The political landscape surrounding cannabis legalization in the US varies across states [3, 4]. Similarly, cannabis use and vaping prevalence also vary widely across states [5, 6]. Although cannabis use is higher in states where cannabis is legalized, evidence on the effect of cannabis legalization on prevalence and modes of cannabis use among youth and adults has been conflicting [7–10]. It is unclear whether EVALI caseload at the state level was positively associated with the state cannabis vaping prevalence after accounting for the state cannabis policies. We hypothesized that cannabis vaping at the state level would be positively associated with the EVALI case burden. Therefore, we examined the cross-sectional association between state-level EVALI caseload and cannabis vaping prevalence in 2019.

## Materials and methods

We used data for the 2019 Behavioral Risk Factor Surveillance System (BRFSS) and the Centers for Disease Control and Prevention (CDC) in this cross-sectional analysis of the association between state-level EVALI caseload and cannabis vaping prevalence. We obtained data on the prevalence of cannabis vaping for each state using the BRFSS and data on the number of EVALI cases, reported as a range, for each state from the CDC [1]. We included data from the 13 states with data on cannabis vaping in the 2019 BRFSS: California, Idaho, Illinois, Maryland, Minnesota, New Hampshire, North Dakota, Oklahoma, South Carolina, Tennessee, Utah, West Virginia, and Wyoming [11].

States were classified as having either recreational, medical, or prohibitive cannabis laws, based on whether a state had recreational or medical laws before or during the EVALI outbreak [1, 3]. Using data from the 2019 BRFSS, we calculated the weighted prevalence of past-month cannabis vaping among non-elderly adults (18–64 years) for each state. Data on the number of EVALI cases, reported as a range, for each state were retrieved from the CDC. We used the midpoint of this range and the 2019 US population estimates for persons aged 13–64 years to generate the number of EVALI cases per million population aged 13–64 years [12]. This age group was used because the age range of reported EVALI cases was 13–85 years, with the majority of cases occurring in non-elderly persons (<65 years) [1, 13].

Linear regression models were used to examine the cross-sectional association between cannabis vaping prevalence and EVALI cases per million population, adjusted for state cannabis legalization policies. Indicator variables were used for medical and recreational cannabis states, leaving prohibitive states as reference. The coefficients obtained from regression models were interpreted as the change in mean EVALI cases per million population for each 1% increase in cannabis vaping prevalence. As a sensitivity analysis, we restricted the calculation of the cannabis vaping prevalence to young adults aged <35 years since about 76% of all EVALI cases were reported in this age group [1].

This work was excluded from review by an institutional review board since it uses publicly available de-identified BRFSS and CDC data. All analyses were conducted using Stata version 16 (StataCorp, College Station, TX). The survey command "svy" was used to account for the complex weighting methodology used by the BRFSS, and a 2-sided alpha (α) level of <0.05 was used to determine statistical significance.

## Results

Of the 13 states in our current study, four were classified as prohibitive cannabis law states (Idaho, South Carolina, Tennessee, and Wyoming), seven as medical cannabis law states

(Maryland, Minnesota, North Dakota, New Hampshire, Oklahoma, Utah, and West Virginia), and two as recreational cannabis law state (California and Illinois). The weighted prevalence of cannabis vaping among those aged 18–64 years ranged from 1.14% (95% CI, 0.61%-2.12%) in Wyoming to 3.11% (95% CI, 2.16%-4.44%) in New Hampshire (**Table 1**). States with prohibitive cannabis laws generally had lower cannabis vaping prevalence (mean: 1.44%) than states with medical (mean: 1.88%) or recreational cannabis laws (mean: 2.29%) (**Fig 1A**). EVALI cases per million population ranged from 1.90 (0.38–3.42) in Oklahoma to 59.10 (19.70–96.53) in North Dakota (**Fig 1B** and **Table 1**).

No significant positive association was observed between cannabis vaping prevalence and EVALI case burden adjusted for state cannabis policies (Coefficient, -18.6; 95%CI, -37.5–0.4; p-value, 0.05). When the cannabis vaping prevalence was restricted to adults aged <35 years, a significant inverse association was obtained; mean EVALI cases per million population decreased by 10.6 cases for each 1% increase in cannabis vaping prevalence (Coefficient, -10.6; 95%CI, -19.9–-1.3; p-value, 0.030; **Table 2**).

## Discussion

Using data from the 2019 BRFSS and the CDC EVALI case reports, we found that states with prohibitive cannabis laws generally had a lower prevalence of cannabis vaping than states with medical or recreational cannabis laws. State-level cannabis vaping prevalence was not positively associated with EVALI caseload, even after accounting for state cannabis policies.

Our finding of an inverse relationship between state-level cannabis vaping prevalence and EVALI caseload is consistent with a prior study by Friedman, which also found that states with higher rates of cannabis use, in general, had lower EVALI prevalence [14]. These findings, therefore, suggest that there may not be a direct, simple link between a state's cannabis vaping prevalence and EVALI cases, but rather the relationship is likely more nuanced, supporting the CDC's hypothesis that the EVALI outbreak is likely reflective of access to informal sources of THC-containing e-liquids [1].

Although cannabis vaping prevalence was low in states with prohibitive cannabis laws, individuals from such states may more likely obtain cannabis from illegal sources, increasing their

**Table 1. Weighted prevalence of cannabis vaping and EVALI cases per million population by state.**

| State | Weighted Prevalence of Cannabis Vaping, % (95% confidence intervals) | | EVALI cases per million population* |
|---|---|---|---|
| | Among persons aged 18–64 years | Among persons aged 18–34 years | |
| California | 2.74 (2.30–3.27) | 3.68 (2.83–4.76) | 6.45 (5.50–7.29) |
| Idaho | 1.20 (0.78–1.83) | 1.77 (0.95–3.27) | 25.47 (8.49–41.60) |
| Illinois | 1.84 (1.39–2.44) | 3.09 (2.10–4.52) | 26.07 (23.17–28.85) |
| Maryland | 1.67 (1.30–2.15) | 3.39 (2.43–4.71) | 18.17 (12.12–23.99) |
| Minnesota | 2.46 (2.08–2.91) | 4.21 (3.35–5.26) | 33.04 (26.44–39.39) |
| New Hampshire | 3.11 (2.16–4.44) | 6.03 (3.80–9.44) | 5.38 (1.08–9.68) |
| North Dakota | 1.20 (0.73–1.98) | 2.22 (1.21–4.04) | 59.10 (19.70–96.53) |
| Oklahoma | 1.94 (1.17–3.18) | 3.37 (1.73–6.44) | 1.90 (0.38–3.42) |
| South Carolina | 1.61 (1.12–2.31) | 2.90 (1.82–4.58) | 8.78 (2.93–14.34) |
| Tennessee | 1.79 (1.15–2.79) | 1.97 (1.00–3.83) | 16.29 (10.86–21.50) |
| Utah | 1.50 (1.19–1.90) | 2.14 (1.56–2.93) | 57.55 (46.04–68.60) |
| West Virginia | 1.26 (0.82–1.94) | 2.07 (1.07–3.97) | 25.64 (8.55–41.88) |
| Wyoming | 1.14 (0.61–2.12) | 2.10 (0.89–4.86) | 13.04 (2.61–23.47) |

*Confidence intervals represent the lower and upper bounds of the range of EVALI cases reported by the CDC (per million population)

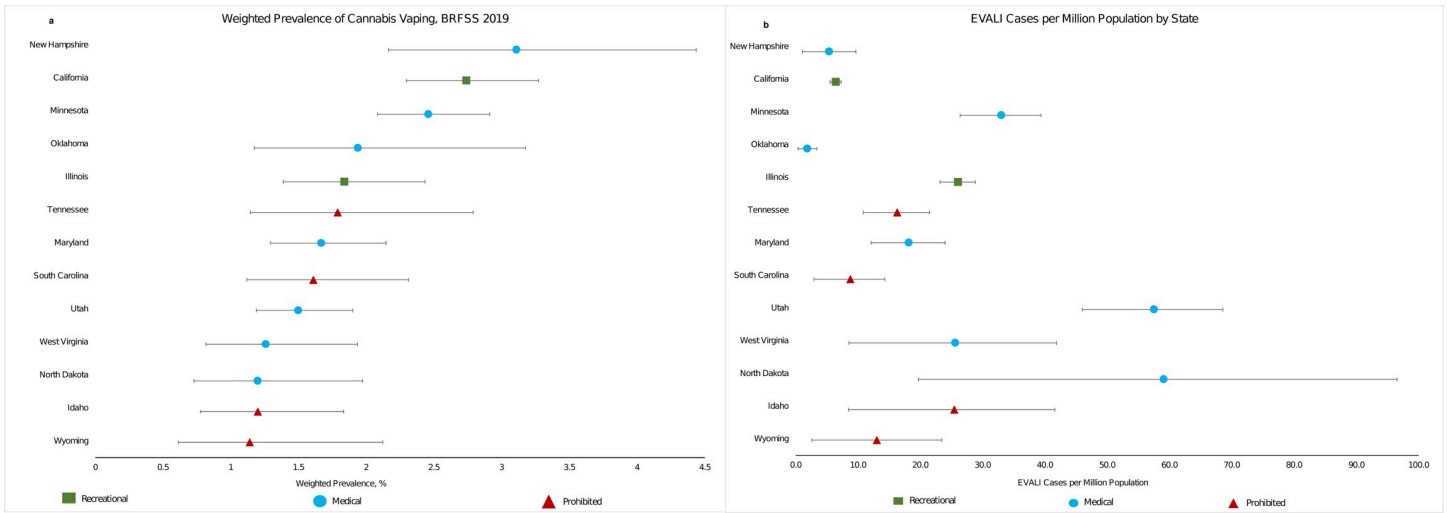

**Fig 1.** a: Cannabis vaping prevalence among persons aged 18–64 years by state, the behavioral risk factor surveillance system, 2019. b: EVALI cases per million population by state.

risk of using contaminated products and hence the higher prevalence of EVALI cases in such states. Conversely, in states with medical or recreational cannabis laws, though with higher cannabis vaping prevalence, individuals are likely to obtain cannabis from legal sources, reducing the risk of contamination. Indeed two recent studies have demonstrated that the presence of legal markets for cannabis may have been protective against EVALI [15, 16].

While the number of reported EVALI cases has significantly declined, continued surveillance of cannabis vaping is warranted. In particular, efforts to discourage black-market sales of contaminated products should be pursued to prevent future outbreaks. The limitations of this study include the small sample size, which may affect the power of our study. The aggregate nature of the data may not reflect observations at the individual level. Also, it is likely that not all EVALI cases were captured during the epidemic; hence, these numbers may underrepresent the true extent of the outbreak. Additionally, the CDC reported the number of EVALI cases as a range, therefore accounting for the wide confidence intervals of the EVALI cases per million population. Finally, there is also the possibility of residual confounding in our analysis of the association between state-level cannabis vaping prevalence and EVALI caseload.

In conclusion, state-level cannabis vaping prevalence was not positively associated with EVALI prevalence. This suggests that the EVALI outbreak may have not necessarily been a simple reflection of state-level cannabis vaping prevalence but rather due to the use of

**Table 2. Table showing the association between state-level cannabis vaping prevalence and EVALI cases per million population.**

| Cannabis Vaping | Coefficient | 95% CI | p-value |
|---|---|---|---|
| Among persons aged 18–64 years | | | |
| Cannabis vaping prevalence | -18.5 | -37.5–0.4 | 0.05 |
| Restricting cannabis vaping prevalence to persons aged 18–34 years | | | |
| Cannabis vaping prevalence | -10.6 | -19.9–-1.3 | 0.03 |

Models adjusted for state cannabis policies (indicator variables used for recreational and medical cannabis states).
CI: Confidence Interval

contaminated or illicitly-sourced vaping products, which are more likely in states with restrictive cannabis laws.

## Author Contributions

**Conceptualization:** Ellen Boakye, Omar El Shahawy, Olufunmilayo Obisesan, Albert D. Osei, John Erhabor, S. M. Iftekhar Uddin, Michael J. Blaha.

**Data curation:** Ellen Boakye.

**Formal analysis:** Ellen Boakye, Omar El Shahawy.

**Funding acquisition:** Michael J. Blaha.

**Methodology:** Ellen Boakye, Omar El Shahawy, Olufunmilayo Obisesan, Albert D. Osei, S. M. Iftekhar Uddin, Michael J. Blaha.

**Supervision:** Michael J. Blaha.

**Visualization:** Omar Dzaye.

**Writing – original draft:** Ellen Boakye.

**Writing – review & editing:** Omar El Shahawy, Olufunmilayo Obisesan, Omar Dzaye, Albert D. Osei, John Erhabor, S. M. Iftekhar Uddin, Michael J. Blaha.

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
