## [Decision Letter · Decision Letter 0]

14 Jul 2022

PONE-D-22-07067The association of state cannabis vaping prevalence with the e-cigarette or vaping product-use associated lung injuryPLOS ONE

Dear Dr. Blaha,

Thank you for submitting your manuscript to PLOS ONE. After careful consideration, we feel that it has merit but does not fully meet PLOS ONE’s publication criteria as it currently stands. Therefore, we invite you to submit a revised version of the manuscript that addresses the points raised during the review process.

We look forward to receiving your revised manuscript.

Kind regards,

Lucy J Troup, Ph.D

Academic Editor

PLOS ONE

Journal Requirements:

Additional Editor Comments:

Thank you so much for your patience with this process. It is getting extremely hard to find reviewers.

Reviewers' comments:

Reviewer's Responses to Questions

**Comments to the Author**

1. Is the manuscript technically sound, and do the data support the conclusions?

Reviewer #1: Partly

Reviewer #2: Yes

2. Has the statistical analysis been performed appropriately and rigorously? 

Reviewer #1: I Don't Know

Reviewer #2: Yes

3. Have the authors made all data underlying the findings in their manuscript fully available?

Reviewer #1: Yes

Reviewer #2: Yes

4. Is the manuscript presented in an intelligible fashion and written in standard English?

Reviewer #1: Yes

Reviewer #2: Yes

5. Review Comments to the Author

Reviewer #1: This is important work and this reviewer sincerely appreciates the undertaking of these authors. There is one important problem with this manuscript which I suspect could be easily resolved by the authors: no method of adjustment of estimates for state cannabis policy category is explained other than the method of categorization itself, which is insufficient to understand the implications of assumptions made in adjustment or the strategy used. If authors would briefly explain their approach to adjustment by category in 1-2 sentences, I suspect it would render this manuscript quite technically sound and would clarify the appropriateness and rigor of the statistical analysis.

Line 2: Given that the investigation ultimately found an inverse association between state cannabis vaping prevalence and EVALI case burden, authors might consider a revised title. (For example, “Inverse association…”)

Line 23: During the EVALI outbreak, cases also occurred among people who were not exposed to THC-containing products, and among those not exposed to VEA. Therefore, authors should strongly consider modifying the word “associated” with a precedent “primarily” or “largely”.

Line 25: VEA was not used in all THC-containing e-liquids. Would strongly recommend adding the word “sometimes” after “additive”.

Line 30: I am not recommending that adjustment is further defined in the abstract. However, it is important that authors describe their method of adjustment in Methods.

Line 47: This sentence is not correct as written. Many persons diagnosed with EVALI had underlying medical conditions. Would recommend either deleting “otherwise healthy”, or moving it to the end of the sentence with a qualifier. (For example: “was an acute lung injury seen in persons who reported e-cigarette use before the onset of symptoms, many of whom were otherwise healthy.”)

Line 63: Materials and Methods is not complete without a description of the method of adjustment for cannabis policy type.

Line 86: Adjusted *how*, though? With what underlying assumptions or interpretation of each category of cannabis policy? This matters very much for the interpretation of results.

Line 89: The inclusion of this sensitivity analysis is a nice addition.

Line 122 (Table 2): This table has a hanging asterisk – which numbers are adjusted for cannabis policy?

Line 129: How did authors account for state cannabis policy? Without explanation of the strategy for adjustment in Methods, one wonders whether these results are a possible artifact of this adjustment.

Line 136: Please consider avoiding use of the phrase "black market". This phrase perpetuates the symbolism of white as "good" and black as "bad" that is unfortunately already quite pervasive.

Line 137: As CDC does not use the phrase “black-market”, this alone is not a correct citation.

Line 149: Authors might consider including in stated limitations the possibility that not all EVALI cases were captured during the epidemic.

Reviewer #2: This was a very well designed study, using existing information on Cannabis use, vaping, Cannabis laws, and EVALI cases. My only suggestion is to add a statement on the large width of several of the confidence intervals in Table 1, page 11, EVALI cases per million population.

6. PLOS authors have the option to publish the peer review history of their article (what does this mean?). If published, this will include your full peer review and any attached files.

Reviewer #1: No

Reviewer #2: **Yes: **Gordon Vrdoljak

---

## [Author Response · Author response to Decision Letter 0]

29 Jul 2022

Reviewer #1: 

This is an important work, and this reviewer sincerely appreciates the undertaking of these authors. There is one important problem with this manuscript which I suspect could be easily resolved by the authors: no method of adjustment of estimates for state cannabis policy category is explained other than the method of categorization itself, which is insufficient to understand the implications of assumptions made in adjustment or the strategy used. If authors would briefly explain their approach to adjustment by category in 1-2 sentences, I suspect it would render this manuscript quite technically sound and would clarify the appropriateness and rigor of the statistical analysis.

We greatly appreciate your thorough review of our manuscript. States were defined by whether they had recreational and/or medical cannabis laws or none. In our regression analysis, we included recreational and medical cannabis states as indicator variables (yes/no), leaving prohibitive states as reference. We have included in the methods section how we defined and modeled state cannabis policy for our analysis.

Lines 83-86

“Linear regression models were used to examine the cross-sectional association between cannabis vaping prevalence and EVALI cases per million population, adjusted for state cannabis legalization policies. Indicator variables were used for medical and recreational cannabis states, leaving prohibitive states as reference.”

Additionally, even when included as a 3-level categorical variable, we obtain the same estimates for the constant and the coefficient of cannabis vaping prevalence. Thank you.

Line 2: Given that the investigation ultimately found an inverse association between state cannabis vaping prevalence and EVALI case burden, authors might consider a revised title. (For example, “Inverse association…”)

Thank you for this suggestion. We have edited the title accordingly. 

“The inverse association of state cannabis vaping prevalence with the e-cigarette or vaping product-use associated lung injury”

Line 23: During the EVALI outbreak, cases also occurred among people who were not exposed to THC-containing products, and among those not exposed to VEA. Therefore, authors should strongly consider modifying the word “associated” with a precedent “primarily” or “largely”.

Line 25: VEA was not used in all THC-containing e-liquids. Would strongly recommend adding the word “sometimes” after “additive”.

We agree that while a majority of hospitalized EVALI patients reported using THC-containing products (82%), there were some who were not exposed to THC-containing e-liquids. We have modified the statement. We have also added “sometimes” after “additive.”

Lines 23-26

“The e-cigarette or vaping product-use-associated lung injury (EVALI) epidemic was primarily associated with the use of e-cigarettes containing tetrahydrocannabinol (THC)- the principal psychoactive substance in cannabis, and vitamin-E-acetate- an additive sometimes used in informally sourced THC-containing e-liquids.”

Line 30: I am not recommending that adjustment is further defined in the abstract. However, it is important that authors describe their method of adjustment in Methods.

Thank you. We have described the method of adjustment under the Methods Section.

Line 47: This sentence is not correct as written. Many persons diagnosed with EVALI had underlying medical conditions. Would recommend either deleting “otherwise healthy” or moving it to the end of the sentence with a qualifier. (For example: “was an acute lung injury seen in persons who reported e-cigarette use before the onset of symptoms, many of whom were otherwise healthy.”)

Thank you for this suggestion as well. We have edited accordingly.

Lines 46-48

“The e-cigarette or vaping product-use-associated lung injury (EVALI) epidemic in the United States (US) was an acute lung injury seen in persons who reported e-cigarette use before the onset of symptoms, many of whom were otherwise healthy”

Line 63: Materials and Methods is not complete without a description of the method of adjustment for cannabis policy type.

Line 86: Adjusted *how*, though? With what underlying assumptions or interpretation of each category of cannabis policy? This matters very much for the interpretation of results.

 Thanks again for this clarifying question. States were defined by whether they had recreational and/or medical cannabis laws or none. In our regression analysis, we included recreational and medical cannabis states as indicator variables (yes/no), leaving prohibitive states as reference.

Lines 83-86

“Linear regression models were used to examine the cross-sectional association between cannabis vaping prevalence and EVALI cases per million population, adjusted for state cannabis legalization policies. Indicator variables were used for medical and recreational cannabis states, leaving prohibitive states as reference.”

Line 89: The inclusion of this sensitivity analysis is a nice addition.

Thanks for acknowledging this.

Line 122 (Table 2): This table has a hanging asterisk – which numbers are adjusted for cannabis policy?

We have deleted the hanging asterisk. The models presented are adjusted for the indicator cannabis policy variables.

Line 129: How did authors account for state cannabis policy? Without explanation of the strategy for adjustment in Methods, one wonders whether these results are a possible artifact of this adjustment.

We have included statements in the methods to clarify this. Please see above. Indicator variables were used to represent recreational and medical states (yes/no). These indicator variables were included in the linear regression models. Thank you.

Line 136: Please consider avoiding use of the phrase "black market". This phrase perpetuates the symbolism of white as "good" and black as "bad" that is unfortunately already quite pervasive. Line 137: As CDC does not use the phrase “black-market”, this alone is not a correct citation.

We appreciate this correction. We have deleted “black market” and kept “informal sources” as used by the CDC.

“These findings, therefore, suggest that there may not be a direct, simple link between a state’s cannabis vaping prevalence and EVALI cases, but rather the relationship is likely more nuanced, supporting the CDC’s hypothesis that the EVALI outbreak is likely reflective of access to informal sources of THC-containing e-liquids”

Line 149: Authors might consider including in stated limitations the possibility that not all EVALI cases were captured during the epidemic.

We have included this as a limitation.

“Additionally, it is likely that not all EVALI cases were captured during the epidemic; hence, these numbers may underrepresent the true extent of the outbreak.” 

Reviewer #2: 

This was a very well-designed study, using existing information on Cannabis use, vaping, Cannabis laws, and EVALI cases. My only suggestion is to add a statement on the large width of several of the confidence intervals in Table 1, page 11, EVALI cases per million population.

Thank you for reviewing our manuscript. We have included this as a limitation and added a footnote to Table 1.

“Additionally, the CDC reported the number of EVALI cases as a range, therefore accounting for the wide confidence intervals of the EVALI cases per million population.”

Editorial Requirements

We have followed all the style requirements in the revised version of our manuscript. Thank you.

All references cited are correct and complete. Thank you.

---

## [Decision Letter · Decision Letter 1]

2 Oct 2022

The inverse association of state cannabis vaping prevalence with the e-cigarette or vaping product-use associated lung injury

PONE-D-22-07067R1

Dear Dr. Blaha,

We’re pleased to inform you that your manuscript has been judged scientifically suitable for publication and will be formally accepted for publication once it meets all outstanding technical requirements.

Kind regards,

Lucy J Troup, Ph.D

Academic Editor

PLOS ONE

Additional Editor Comments (optional):

Reviewers' comments:

Reviewer's Responses to Questions

**Comments to the Author**

1. If the authors have adequately addressed your comments raised in a previous round of review and you feel that this manuscript is now acceptable for publication, you may indicate that here to bypass the “Comments to the Author” section, enter your conflict of interest statement in the “Confidential to Editor” section, and submit your "Accept" recommendation.

Reviewer #1: All comments have been addressed

Reviewer #2: All comments have been addressed

2. Is the manuscript technically sound, and do the data support the conclusions?

Reviewer #1: Yes

Reviewer #2: Yes

3. Has the statistical analysis been performed appropriately and rigorously? 

Reviewer #1: Yes

Reviewer #2: Yes

4. Have the authors made all data underlying the findings in their manuscript fully available?

Reviewer #1: Yes

Reviewer #2: Yes

5. Is the manuscript presented in an intelligible fashion and written in standard English?

Reviewer #1: Yes

Reviewer #2: Yes

6. Review Comments to the Author

Reviewer #1: Thank you for the thorough explanations and the added content to Methods -- this addresses all the previous concerns of this reviewer.

Reviewer #2: Thank you for addressing all the comments posed by all of the reviewers. The paper is greatly improved.

7. PLOS authors have the option to publish the peer review history of their article (what does this mean?). If published, this will include your full peer review and any attached files.

Reviewer #1: No

Reviewer #2: **Yes: **Gordon Vrdoljak

---

## [Editor Report · Acceptance letter]

6 Oct 2022

PONE-D-22-07067R1 

The inverse association of state cannabis vaping prevalence with the e-cigarette or vaping product-use associated lung injury 

Dear Dr. Blaha:

I'm pleased to inform you that your manuscript has been deemed suitable for publication in PLOS ONE. Congratulations! Your manuscript is now with our production department. 

Kind regards, 

on behalf of

Dr. Lucy J Troup 

Academic Editor

PLOS ONE